# Controversy and Consideration of Refractive Surgery in Patients with Heritable Disorders of Connective Tissue

**DOI:** 10.3390/jcm10173769

**Published:** 2021-08-24

**Authors:** Majid Moshirfar, Matthew R. Barke, Rachel Huynh, Austin J. Waite, Briana Ply, Yasmyne C. Ronquillo, Phillip C. Hoopes

**Affiliations:** 1Hoopes Vision Research Center, Hoopes Vision, Draper, UT 84020, USA; bply@hoopesvision.com (B.P.); yronquillo@hoopesvision.com (Y.C.R.); pch@hoopesvision.com (P.C.H.); 2John A. Moran Eye Center, Department of Ophthalmology and Visual Sciences, University of Utah, Salt Lake City, UT 84132, USA; 3Utah Lions Eye Bank, Murray, UT 84107, USA; 4McGovern Medical School at the University of Texas Health Science Center, Houston, TX 77030, USA; matthew.r.barke@uth.tmc.edu; 5University of Utah School of Medicine, Salt Lake City, UT 84132, USA; Rachel.huynh@hsc.utah.edu; 6A.T. Still University College of Osteopathic Medicine in Arizona, Mesa, AZ 85206, USA; austinjwaite@atsu.edu

**Keywords:** LASIK, PRK, refractive surgery, osteogenesis imperfecta, ehlers danlos syndrome, marfan syndrome, loeys-dietz syndrome, epidermolysis bullosa, stickler syndrome, wagner syndrome, pseudoxanthoma elasticum

## Abstract

Heritable Disorders of Connective Tissue (HDCTs) are syndromes that disrupt connective tissue integrity. They include Osteogenesis Imperfecta (OI), Ehlers Danlos Syndrome (EDS), Marfan Syndrome (MFS), Loeys-Dietz Syndrome (LDS), Epidermolysis Bullosa (EB), Stickler Syndrome (STL), Wagner Syndrome, and Pseudoxanthoma Elasticum (PXE). Because many patients with HDCTs have ocular symptoms, commonly myopia, they will often present to the clinic seeking refractive surgery. Currently, corrective measures are limited, as the FDA contraindicates laser-assisted in-situ keratomileusis (LASIK) in EDS and discourages the procedure in OI and MFS due to a theoretically increased risk of post-LASIK ectasia, poor wound healing, poor refractive predictability, underlying keratoconus, and globe rupture. While these disorders present with a wide range of ocular manifestations that are associated with an increased risk of post-LASIK complications (e.g., thinned corneas, ocular fragility, keratoconus, glaucoma, ectopia lentis, retinal detachment, angioid streaks, and ocular surface disease), their occurrence and severity are highly variable among patients. Therefore, an HDCT diagnosis should not warrant an immediate disqualification for refractive surgery. Patients with minimal ocular manifestations can consider LASIK. In contrast, those with preoperative signs of corneal thinning and ocular fragility may find the combination of collagen cross-linking (CXL) with either photorefractive keratotomy (PRK), small incision lenticule extraction (SMILE) or a phakic intraocular lens (pIOL) implant to be more suitable options. However, evidence of refractive surgery performed on patients with HDCTs is limited, and surgeons must fully inform patients of the unknown risks and complications before proceeding. This paper serves as a guideline for future studies to evaluate refractive surgery outcomes in patients with HDCTs.

## 1. Introduction

Heritable Disorders of Connective Tissue (HDCTs) are a group of syndromes that disrupt connective tissue integrity and often cause systemic manifestations. HDCTs involving ocular manifestations include Osteogenesis Imperfecta (OI), Ehlers-Danlos Syndrome (EDS), Marfan Syndrome (MFS), Loeys-Dietz Syndrome (LDS), and Epidermolysis Bullosa (EB), Stickler Syndrome (STL), Wagner Syndrome, and Pseudoxanthoma Elasticum (PXE). Myopia is a common issue in patients with HDCTs and inevitably leads to patients seeking refractive surgery consultations.

Currently, corrective measures are limited, as the FDA states laser-assisted in-situ keratomileusis (LASIK) is an absolute contraindication in EDS. It is not recommended in disorders with abnormal collagen (e.g., MFS and OI) due to a theoretical increased risk of post-LASIK ectasia, poor wound healing, poor refractive predictability, and globe rupture. The concern for post-LASIK ectasia is based on the Ectasia risk score system, which lists abnormal preoperative corneal topography, low residual stromal bed thickness, young age, and thin preoperative corneal thickness as common risk factors in order of significance [1]. Another risk factor is the biomechanical weakening of the cornea from lower corneal hysteresis (CH), which is associated with a thinner central corneal thickness (CCT) and increased intraocular pressure (IOP) [2]. Due to these risk factors, there is potential for corneal or scleral rupture or staphyloma even without refractive surgeries being performed. Many of these risk factors (e.g., low CCT, keratoconus, increased rate of global ruptures) are commonly associated with the various HDCTs.

However, the above risk factors may not be problematic for every person with an HDCT diagnosis. For example, reduced CCT is associated with specific gene sequence variants [3], and clinical presentation for each HDCT is variable and wide-ranging in symptom severity [4,5,6,7,8,9,10,11,12,13,14,15,16,17,18]. The exact rate of post-LASIK complications has been difficult to assess due to limited refractive surgery cases performed on those diagnosed with HDCTs. A literature search revealed only one study surveying patients with EDS about their ophthalmic surgical experiences, showing 43% of patients had undergone radial keratotomy, PRK, LASIK, or LASEK. Of those, 23.3% reported complications, including under correction/regression (18.6%), postoperative pain (9.3%), impaired night vision (7%), dry eye (4.7%), induced astigmatism (7%) and corneal ectasia (4.7%) [19]. The survey demonstrates that refractive surgery on an EDS patient does not make any complication a foregone conclusion. A current perspective on ocular management of patients with MFS suggests corneal refractive surgery can be performed in those without lens dislocation and with mild cases of myopia [8].

Furthermore, the risk of post-LASIK ectasia continues to decrease due to advances in LASIK surgery and preoperative risk analysis [20]. Screening for refractive surgery eligibility is better due to advancements in measuring corneal tomography and hysteresis. These technological changes allow proper evaluation for the risk of ectasia in those diagnosed with HDCTs. Thus, those affected less symptomatically may undergo refractive surgery with safe outcomes.

Overall, the clinical variability and improvements in preoperative screening raise the question of whether a blanket contraindication to refractive surgery in patients with HDCTs is appropriate. This paper details the various clinical presentations of OI, EDS, MFS, LDS, EB, STL, Wagner Syndrome, and PXE and their subtypes to evaluate the spectrum of possibilities for refractive surgery. It also expands on the ocular manifestations that require consideration and evaluation preceding refractive surgery. Finally, it provides a framework to approach the therapeutic possibilities for refractive error correction in each HDCT.

## 2. Osteogenesis Imperfecta

Osteogenesis Imperfecta (OI) is a disorder disrupting type I collagen, affecting around 1 in 15,000 births and an estimated 25–50,000 people in the United States [21,22]. Approximately 90% of cases are autosomal dominant (AD) inherited mutations in COL1A1 or COL1A2 [4]. Because type I collagen contributes to tensile strength in tissue, common manifestations of OI are long bone fractures, low bone mineral density, bone pain, hearing loss, blue sclera, joint laxity, scoliosis, dental abnormalities, subcutaneous hemorrhages, and heart and lung problems [4,21]. However, symptoms vary between the five classifications of OI: type 1 typically presents with blue sclera and a mild, non-deforming phenotype due to a quantitative defect; type 2 is severe and lethal perinatally; type 3 presents with blue sclera and a moderate to severe, progressively deforming phenotype resulting in multiple fractures; type 4 is moderate with normal sclera and limited fractures; type 5 involves calcification of interosseous membranes (Table 1) [4,5,23].

While type I collagen is found throughout the eye, it is primarily seen in the cornea and sclera. Type 1 collagen makes up around 70% of the cornea [26] and 90% of the sclera [5]. The absence of corneal K-structures, a sub-Bowman’s fibrous structure, in OI can lead to an absent or atrophic Bowman’s layer maybe be another reason for corneal stromal thinning [27]. The scleral integrity can be severely disrupted and translucent to the underlying uvea, resulting in blue sclera. OI can also present as thin cornea, megalocornea, keratoconus, ocular fragility, zonular cataracts, dislocated lens, congenital glaucoma, optic atrophy, papilledema, partial color blindness, detachment of Descemet’s membrane, or retinal and subhyaloid hemorrhage [21,28,29]. Eye rubbing or finger trauma can lead to corneal or global rupture due to increased ocular fragility, creating concern for scleral perforation during routine procedures [5].

Studies have routinely found patients with OI have a thinner CCT that ranges from 362–571 µm, with 52.9% below 500 µm [21]. A study revealed patients with blue sclera or type 1 OI have a significantly lower CCT [24,30,31]. The average CCT for types 3 and 4 is higher, at 510 µm and 500 µm, respectively [30]. Additionally, reduced CH has been observed in children with OI [24], though keratoconus has not been frequently seen [32].

## 3. Ehlers Danlos Syndrome

Ehlers Danlos Syndrome (EDS) is a heterogeneous group of HCTDs sharing characteristic features of joint hypermobility, skin hyperextensibility, and tissue fragility due to abnormal type 5 collagen [7]. The prevalence is ~ 1 in 5000 [33], with no specific inheritance pattern. It can be either AD or autosomal recessive (AR), depending on the subtype. Currently, the 13 subtypes of EDS are classified based on their varying clinical presentations: classical, classical-like, cardiac-valvular, vascular, hypermobile, arthrochalasia, dermatosparaxis, kyphoscoliotic, spondylodysplastic, musculocontractural, myopathic, periodontal, and Brittle Cornea Syndrome (BCS) [7].

The ocular manifestations within EDS are wide-ranging, and their prevalence and severity differ among subtypes (Table 2). Patients with EDS can present with blue sclera, epicanthic folds, floppy eyelids, widely spaced eyes, strabismus, high myopia with retinal detachment, keratoglobus/keratoconus, dry eyes, corneal fragility, and angioid streaks [17,34]. Classical, Kyphoscoliotic, and BCS subtypes are associated with significant ophthalmologic findings (i.e., globe rupture) [35,36]. The other subtypes present with minor ophthalmologic findings, except myopathic and periodontal EDS, with no reported eye findings [6].

Classical EDS’s more severe ocular manifestations include blue sclera, thin CCT (410–450 µm), and steep corneas but without a known predisposition to keratoconus [26,41,42]. A case study of 62 patients with classical EDS found blue sclera in 84% of patients [43]. Conjunctivochalasis has also been reported [37]. The kyphoscoliotic subtype has frequent occurrences of corneal rupture with minimal trauma frequently occurs, and corneal pathology has shown an absent Bowman’s layer, marked stromal thinning, and Descemet’s membrane abnormalities [39]. Blue sclera, microcornea, corneal thinning (CCT as low as 400 µm), keratoconus, and keratoglobus may also be present [39,44]. BCS also has a high risk of corneal rupture but is less frequent than kyphoscoliotic EDS [45]. However, their CCT can be thinner than kyphoscoliotic EDS, reaching as low as 200 µm. BCS is also associated with stromal thinning, myopia, blue sclera, keratoconus, keratoglobus, and megalocornea [40,46].

For the less severe ocular findings, hypermobile EDS may present with dry eyes, pathologic myopia, vitreous abnormalities, and asymptomatic lens opacities [38]. In a study of 44 eyes with hypermobile EDS, no cases of keratoconus or significant differences in CCT (average CCT 540s µm) were found compared to controls [38,41]. However, corneal epithelial density was significantly lower, and stromal keratocyte density was higher [38]. In vascular EDS, a common feature is subtle globe protrusion, but retinal disorders, increased risk of globe rupture, and keratoconus are not common [41,46]. Cardiac-valvular, arthrochalasia, dermatosparaxis, and musculocontractural can present with myopia, astigmatism, and blue sclera [6]. Hypermetropia, microcornea, and corneal clouding have been reported in spondylodysplastic EDS [6]. Additionally, classical-like EDS has been reported with recurrent subconjunctival hemorrhages [47].

## 4. Marfan Syndrome

Marfan syndrome (MFS) is an AD inherited mutation in the FBN1 gene on chromosome 15 that encodes for fibrillin-147, leading to excessive signaling and activation of TBF-beta [9]. The prevalence is 1 in 5000 to 10,000 people [48]. The main clinical manifestations are long bone overgrowth, aortic root aneurysm, and ectopia lentis. Other common features include hypermobility, low bone mineral density, scoliosis, pectus excavatum and carinatum, dural ectasia, foot deformities, and generalized ligamentous laxity (Table 3) [9].

The most common ocular abnormality is ectopia lentis, occurring in 60–80% of cases due to the presence of fibrillin-1 in ciliary zonules [8]. Their corneas are also more deformed due to decreased bending resistance and capacity to dissipate energy [58,59]. Other associated findings include myopia, flat cornea, astigmatism, thinner CCT (thinned CCT as low as 502 µm), premature cataracts, retinal detachment (5–25.6%), glaucoma (33%), and anisocoria [8,49,60,61]. The prevalence of myopia is between 33–63%, with over 50% of those affected having ≥ –3D of myopia [8,62,63]. On the other hand, Konradsen found 61% of patients had < –3D of refractive error and flatter corneas, which compensated for increased myopia [64]. Children with MFS are more myopic and have decreased corneal curvature, CCT, and best-corrected visual acuity (BCVA) than controls [65].

## 5. Loeys-Dietz Syndrome

Loeys-Dietz Syndrome is an AD inherited disorder with mutations in TGF beta receptor 1 (TGFBR1), TGF beta receptor 2 (TGFBR2), TGF beta 2 (TGFB2), TGF beta 3 (TGFB3), or SMAD3 [10,50,66], with an estimated prevalence of ≤1 in 100,000 [51]. Systemic manifestations of LDS are similar to MFS, including vascular aneurysms (cerebral, thoracic, or abdominal), skeletal abnormalities (pectus excavatum or carinatum), scoliosis, joint laxity, and craniofacial abnormalities (cleft palate, craniosynostosis, or bifid uvula) [10]. Allergies, skin abnormalities, neurological findings, pulmonary manifestations, and pregnancy-related changes have also been reported [67,68]. The distinguishing characteristic separating LDS from MFS is the lack of lens dislocation [69].

Ocular manifestations include myopia, blue or dusky sclera, cataract, retinal detachment, retinal tortuosity, strabismus, and amblyopia [50,68,69,70]. An initial study involving 14 patients with LDS noted 13 with hypertelorism, seven with exotropia, and eight with blue sclera [50]. However, another retrospective review found no patients with hypertelorism or blue/dusky sclera [10], making it challenging to confirm these findings as diagnostic criteria for LDS. Patients with LDS have decreased CCT and increased myopia rates compared to controls, though myopia was less common and severe compared to patients with MFS [10]. The study found CCT was 521 +/− 48 µm in LDS compared to 542 +/− 37 µm in controls [10]. Further, presentations vary between genotypes, with more pronounced myopia, decreased CCT, and increased interpupillary distance (in men) in TGFBR2 compared to TGFBR1 [10].

## 6. Epidermolysis Bullosa

Epidermolysis Bullosa (EB) is characterized by epithelial tissue fragility, resulting in blisters and erosions from minimal trauma [71]. There are four main types of inherited EB: EB simplex (EBS), junctional EB (JEB), dystrophic EB (DEB), and Kindler Syndrome [17]. The prevalence of EB in the US is 0.4–4.6 per million and, specifically, 0.36 per million people for DEB [12]. EB can be inherited in an AD or AR form with mutations in keratin, laminin, collagen, or kindlin [71]. Dominantly inherited DEB (DDEB) presents at birth with skin blistering, dermal scarring, milia, and dystrophic nails, with normal teeth and oral mucosa. Recessively inherited DEB (RDEB) presents at birth with a wider range of symptoms, such as dermal blisters on the knees and elbows leading to joint deformity, polysyndactyly from scarring around the fingers, and oral mucosa involvement with tooth decay [11].

Eye involvement is common in EB, particularly the surface. Short-lived manifestations may include tearing, blistering of the eyelids, corneal erosions, conjunctival injection, and bullous keratopathy (Table 3) [17]. The presence of corneal erosions and blisters are high in RDEB (74.1%) and JEB (47.5%) and less prevalent in DDEB (2.12%) and EBS (6.19%) [12]. These issues can be treated with artificial tears, topical antibiotics, Vitamin A, topical fibronectin, and soft contact lenses [72]. A reduced tear break-up time (<8 s) occurs in 95.1% and an abnormal Schirmer test (<15 mm) in 92.4% [73], signifying dry eye disease (DED). Prolonged DED can lead to low corneal sensitivity, decreased cellular cohesion, poor tear quality, squamous metaplasia of the conjunctiva, and goblet cell loss [72], and requires treatment with lubricants/artificial tears [74].

Chronic sequelae such as pannus formation, corneal scarring, symblepharon, ankyloblepharon, and ectropion can occur and lead to significant vision loss or blindness [17]. Corneal scarring occurred in 50% of RDEB and 26.83% of JEB patients, but only in 0.95% of DDEB and 3.16% of EBS patients [12]. In RDEB, symblepharon and blepharitis were common findings, with 10.07% and 17.52% of patients affected, respectively [12]. Rare presentations include amblyopia, cataracts, strabismus, pseudopterygia, and microphthalmos [74].

## 7. Stickler Syndrome

Stickler Syndrome (STL) presents with conductive and sensorineural hearing loss, midfacial underdevelopment, cleft palate, and spondyloepiphyseal dysplasia [52,75]. It has an estimated prevalence of 1–3 in 10,000 [53]. This disorder can be caused by either AD inherited mutations of collagen type 2, collagen type 11, and lysyl oxidase or by AR inherited mutations in collagen type 9 and lysyl oxidase [13]. The most commonly seen mutations are in COL2A1 (STL type 1) and COL11A1 (STL type 2) [13].

Ocular findings include myopia, cataracts, vitreous alterations, glaucoma, and retinal detachments (Table 3) [52,75]. Huang found that 76% had high myopia (>−6D), and 69% had retinal detachment, many of which had a COL2A1 mutation [13]. One study found in COL2A1 mutations that 89% had myopia, 42% had vitreous abnormalities, and 55% had at least one retinal detachment [52]. Mutations in COL2A1 and COL11A1 are associated with early-onset high myopia [76]; however, myopia of ≥−10D was more common in COL2A1 than in COL11A1 (40% vs. 19%) [14]. Additionally, cataracts are more common in COL11A1 than in COL2A1 (59% vs. 36%) [14]. Studies show high rates of retinal detachment (45–69%) [13,14], causing poor visual acuity. Approximately 60–70% of individuals with STL type 1% and 40% with STL type 2 experience a retinal detachment, usually between 10–30 years of age [77]. Conversely, COL9A1 and COL9A3 rarely presented with retinal detachment [13].

## 8. Wagner Syndrome

Wagner Syndrome is caused by an AD inherited mutation in the VCAN gene on chromosome 5q [54], encoding for versican, an extracellular matrix proteoglycan contributing to the structural integrity of the vitreous [15]. The prevalence is unknown, with an estimated total of 300 individuals affected [55]. The hallmark of Wagner Syndrome is an optically empty vitreous with avascular strands, membranes, or veils [16,54,78]. Other ocular features are myopia, night blindness from chorioretinal atrophy, presenile cataract, retinal detachment, and occasional uveitis beginning in adolescence (Table 3) [78]. In contrast to STL, no systemic abnormalities have been described.

Mild to high myopia and astigmatism are prevalent, with some patients reaching > −10D [15,54,79]. Congenital glaucoma occurs, likely due to altered versican expression during trabecular meshwork development, and often requires surgical intervention [15,16]. Graeminger found peripheral tractional detachments in 55% of the eyes in patients over the age of 45 [16]. Additionally, all patients older than 45 exhibited chorioretinal atrophy and cataracts [16]. Not to mention, chorioretinal abnormalities and retinal detachments can still occur at a young age (5–15 years old) [54]. For this reason, annual visits with a retinal specialist are recommended.

## 9. Pseudoxanthoma Elasticum

Pseudoxanthoma elasticum (PXE) is an AR inherited mutation in the ABCC6 gene on chromosome 6 with a prevalence estimated between 1 in 25,000 to 100,000 people [17,56]. The mutation causes a defective cell membrane transporter with a subsequent buildup of dystrophic calcification in the elastic tissues of the skin, vasculature, and Bruch’s membrane (Table 3) [56,57]. It classically presents with characteristic small yellow papules on the neck and flexural areas that progress into reticulated plaques and cause loose and wrinkly skin. Cardiovascular manifestations include angina pectoris, arterial hypertension, atherosclerosis, valvular disease, myocardial infarction, cerebrovascular accident, and sudden cardiac death [17,56].

The presentation of PXE can be variable. AD type 1 typically has thin and delicate skin, accelerated atherosclerosis with mitral valve disease, and angioid streaks with choroidal neovascularization. AD type 2 has yellow, flat skin papules, skin hyperelasticity, and angioid streaks with blue sclera. AR-type 1, the most common form of PXE, has similar skin lesions and angioid streaks to AD type 1 with added gastrointestinal bleeding. Finally, AR-type 2, the rarest form of PXE, has severe skin manifestations and angioid streaks without other systemic manifestations [80]. PXE should not be confused with juvenile xanthogranuloma, a non-Langerhans cell histiocytosis with yellow or erythematous skin nodules found commonly on the head and neck and tumors in the iris or conjunctiva that may lead to glaucoma, hyphema, or vision loss [81].

The ocular manifestations of PXE primarily involve the posterior segment of the eye, with the most common being angioid streaks (85%) [17]. PXE can also have a peau d’orange appearance on fundoscopy, optic disc drusen (6–20%), chorioretinal atrophy with comet tail lesions, and macular degeneration with hemorrhage [18,82]. The earliest finding is usually the peau d’orange appearance, presenting as spotted hypo- and hyper-fluorescence on microscopy [18,56,82]. Before the age of 15, the peau d’orange appearance is frequently seen without signs of angioid streaks [83]. However, angioid streaks and choroidal neovascularization are typically present by their 40s [84], resulting in visual acuity of 20/200 or worse by their 4th or 5th decade of life [18,80,82]. 

## 10. Refractive Surgery Considerations and Consultation

An HDCT diagnosis is currently viewed as a contraindication to refractive surgery, with surgery attempts to be avoided in these patients. While some patients may remain poor refractive surgery candidates due to the combination of corneal biomechanics and ocular manifestations seen, we believe there is an opportunity to approach refractive surgery in those who have stable refractions and are less severely affected. We suggest a general framework to guide clinicians regarding refractive surgery in patients with HDCTs (Figure 1).

In general, patients with a normal corneal tomography (no signs of keratoconus, asymmetric astigmatism, and pellucid marginal degeneration, a CCT > 500 µm, and white sclera without uveal showing can proceed with any corneal refractive surgery option per patient or physician preference. This assumes a residual stromal bed thickness (RSB) > 300 µm for PRK and LASIK or RSB > 280 µm for SMILE, and a percentage tissue altered (PTA) <40%, as well as a biomechanical evaluation measuring CH and a corneal resistance factor (CRF). An RSB > 280 µm for SMILE can be recommended due to the residual intact anterior corneal cap’s contribution to biomechanical stability, allowing for a reduced RSB in SMILE.

Corneal cross-linking (CXL) immediately after LASIK and the use of femtosecond lasers producing the lowest suction increase in IOP, such as VisuMax (Carl Zeiss Meditec AG, Jena, Germany) [85,86,87], can be utilized as added measures to prevent the development of ectasia or other intraoperative and postoperative complications [88,89]. If VisuMax is unavailable, other options to be considered are the Wavelight FS200 (Alcon Laboratories Inc., Fort Worth, TX, USA), LenSx (Alcon LenSx Inc., Aliso Viejo, CA, USA), or Victus (Bausch & Lomb Incorporated, Rochester, NY, USA). With any refractive surgery, larger optical zones require more tissue removal to achieve the same refractive power, leading to a smaller residual stromal bed [90]. This may increase the risk of ectasia [91] and forward displacement of the posterior cornea [92] and increase ocular fragility. For a 0.5 mm difference in optic zone size, an additional 3−4 microns of tissue is ablated for every diopter of myopic correction [90]. Therefore, using smaller optical zones is recommended. A refractive lens exchange is an option for patients over 55 years old or with early presence of nuclear sclerotic cataract. However, there is loss of accommodation and cumulative risk of retinal detachment over time.

Patients with blue sclera should proceed with caution regarding refractive surgery involving suction, such as LASIK or SMILE, due to the risk of intraoperative scleral rupture. In conjunction, femtosecond platforms with the lowest suction IOP increase (e.g., VisuMax) should be used. Patients who meet the above criteria but have high myopia > −7D may consider proceeding with pIOLs. Those who do not meet an RSB > 300 µm for PRK and LASIK or RSB > 280 µm for SMILE, or a percentage tissue altered (PTA) < 40% may also consider pIOLs. The pIOL can be placed in the posterior chamber, such as the Visian Implantable Collamer Lens (STAAR Surgical Co, Monrovia, CA, USA), or fixated to the iris, such as the Verisyse phakic lens (Artisan; Advanced Medical Optics, Santa Ana, CA, USA). Both options are safe, effective, and predictable in patients with high myopia and astigmatism [93,94,95,96,97,98]. Specifically, those with an increased risk of ectopia lentis, like MFS, should preferably use an iris-fixated pIOL. Those without risk of ectopia lentis, like STL, can proceed with either an iris-fixated or posterior chamber pIOL.

The following treatment considerations are for patients with a normal corneal tomography and biomechanical evaluation but a CCT ≤ 500 µm. For those with a CCT ≤ 400 µm, refractive surgery should be contraindicated regardless of white or blue sclera. For those with CCT > 400 µm, initial CXL can be performed. Patients may proceed with additional refractive surgery if stable refraction and normal tomography are seen 3–12 months after CXL. In patients with white sclera, PRK can be done with an RSB > 300 µm and SMILE with RSB > 280 µm, as long as PTA < 40%. Combining SMILE and CXL has been shown to safely protect against the development of ectasia [99,100]. However, for those with RSB ≤ 280 µm, patients may proceed with a pIOL. On the other hand, in those with blue sclera, PRK is preferred due to no risk of scleral rupture and can be done if the RSB is >300 µm. Those with a lower RSB can proceed with a pIOL.

Lastly, the following options are for patients with an abnormal corneal tomography (Figure 1). Surgery would be contraindicated if patients additionally have a CCT ≤ 400 µm, blue sclera, or ocular fragility (reflected by abnormal indices such as CH and CRF). These cases will likely include patients with BCS, kyphoscoliotic EDS, or OI type 1. Patients with a CCT > 400 µm, biomechanical evaluation, and white sclera can proceed with CXL. The increased corneal stiffness (reflected by improved CH and CRF) in CXL provides additional stability for patients with keratoconus [101,102,103]. However, further research addressing safety is needed as corneal melting has occurred in patients with keratoconus receiving CXL [104]. Patients with stable refraction and normal tomography may proceed with PRK if RSB > 300 µm or SMILE if RSB > 280 µm, as long as PTA < 40%. If the RSB is too low for PRK or SMILE, they can proceed with pIOL, which is effective and safe without keratoconus progression [105,106]. Those with low myopia and early signs of stable keratoconus who received PRK and CXL have been shown to stop disease progression and improve vision [107]. Patients who do not have stable refraction or normal tomography after CXL would be ineligible for refractive surgery.

## 11. Specific Recommendations

In OI patients presenting with blue sclera and very thin CCT (i.e., OI type 1), refractive surgery is likely contraindicated. However, OI type 1 with a CCT > 400 µm may potentially undergo initial CXL. With the presence of thicker corneas (CCT > 500 µm), OI types 3, 4, and 5 are likely eligible for any refractive surgery type, though some may need initial CXL. Given the prevalence of ocular fragility and reduced CH and CRF in patients with OI, femtosecond platforms with the lowest IOP suction increase (e.g., VisuMax) minimize the risk of complications in LASIK or SMILE. Further research is needed to determine the full scope of ocular manifestations and the natural progression of changes in visual acuity.

For kyphoscoliotic EDS and BCS, refractive surgery should not be pursued due to the extreme corneal thinning, presence of keratoconus and keratoglobus, and ocular fragility seen in these patients. Even CXL carries extreme risk as corneal perforation has occurred post-CXL in patients with BCS [36]. Those with normal corneal tomography and a CCT > 500 µm (typically classical-like, cardiac-valvular, arthrochalasia, spondylodysplastic, musculocontractural, myopathic, and periodontal) may receive refractive surgery of their choice. In dermatosparaxis EDS, the presence of congenital or early progressive myopia may make pIOL a more suitable option. Classical EDS often presents with a decreased CCT and steep cornea, making initial CXL the preferred pathway. Vascular EDS should also proceed down this same pathway, as post-LASIK development of myopic regression, Salzmann nodular degeneration, and dry eye syndrome has occurred [108].

Individuals with MFS typically present with normal corneal tomography and CCT > 500 µm. Additionally, a 10-year follow-up of patients with MFS showed stable myopia and no change in the frequency of those with refraction > -3D increased corneal thinning or keratoconus [109]. Given their ocular stability, many patients would be good candidates for any type of refractive surgery. Of note, a subset of these patients can present with high myopia, which would make iris-fixated pIOLs a suitable option in those without significant crystalline lenses or signs of iridodonesis. Patients that present with iridodonesis may benefit from sutureless methods such as the glued technique or Yamane technique. Placement of a posterior segment pIOL in one MFS patient with high myopic astigmatism and lens coloboma showed promising results, with a postoperative bilateral UCVA of 20/20 [110]. However, due to zonular weakness and erosion in these patients, an iris-fixated pIOL is preferred.

The two most common forms of LDS (TGFBR1 and TGFBR2) have average CCTs > 500 µm, and therefore, most are conducive to refractive surgery. With the presence of blue or dusky sclera, PRK would be preferable. An alternative would be LASIK or SMILE using femtosecond platforms with the lowest IOP suction increase (e.g., VisuMax). Keratoconus has been observed in a patient with LDS [10], although more research is needed to determine if initial CXL would benefit these patients.

Regardless of the EB subtype, the ocular surface needs to be optimized before any refractive surgery. It is also necessary for clinicians to evaluate limbal stem cell and corneal epithelium health with impression cytology or high-resolution anterior ocular coherence tomography. The presence of irregular ocular surface and DED in patients with EB makes LASIK a poor choice since dry eyes are the most common post-LASIK complication [111]. Therefore, PRK and SMILE would be more preferable options. The lack of flap in SMILE provides increased corneal stability and decreased incidence of postoperative dry eye compared to LASIK [112]. Patients who cannot optimize their ocular surface or are concerned about the recurrent corneal disease during the healing period of refractive surgery may want to consider pIOL.

Patients with STL present with myopia in childhood, but most studies show it is stable rather than progressive [14], making them promising candidates for surgical correction. However, before any refractive surgery, consultation with a retinal specialist is necessary due to the high prevalence of retinal detachment. No studies have reported their average CCT, but for those with >500 µm, any type of refractive surgery can be performed. Many of these patients will need a pIOL due to the increased prevalence of high myopia. The use of pIOLs needs to be approached with precaution as retinal detachments have occurred in 4.8% and 2.07% of the anterior chamber and posterior chamber pIOL surgeries, respectively [113,114]. Patients presenting with STL type 2 are likely better candidates for refractive surgery than STL type 1 due to decreased retinal detachments and the lower degree of myopia.

Due to the characteristic empty vitreous appearance and other posterior segment abnormalities in Wagner Syndrome, consultation with a vitreoretinal specialist is recommended before refractive surgery. If the vitreoretinal specialist approves, the patient can proceed with any refractive surgical option as their anterior segment is largely unaffected. For patients with high myopia, pIOL is an option but needs to be used cautiously due to the chance of retinal detachments [113,114].

Finally, patients with PXE should consult with a retinal specialist before refractive surgery due to their retinal abnormalities. Because no corneal abnormalities are noted in these patients, all refractive surgical options can be available. However, femtosecond platforms with the lowest IOP suction increase (e.g., VisuMax) may be beneficial in reducing the possibility of subretinal hemorrhage through breaks in Bruch’s membrane. The patient and surgeon should also discuss the long-term outcome of refractive surgery, as it will not solve the underlying retinal issues (i.e., angioid streaks and choroidal neovascularization) responsible for much of the vision loss in patients with PXE. Patients should be aware of the risk of blindness from the natural course of the disease.

## 12. Conclusions

This paper examined several HDCTs to determine whether a blanket contraindication for refractive surgery is appropriate. As myopia is more prevalent among patients with HDCTs, they will inevitably present for a refractive surgery consultation. Based on the ocular manifestations of each HDCT, an all-inclusive absolute contraindication may not be in the best interest of the patient seeking a refractive procedure. Certain HDCTs and their subtypes are more amenable to refractive surgery than others. Because of phenotypic variation, every patient should be evaluated on an individual basis and provided appropriate options. However, surgery should not be performed if these patients do not meet the minimum recommendations for each procedure listed in our framework. Furthermore, any patient presenting with symptoms of HDCTs may be advised for genetic testing with a geneticist who specializes in hypermobility and connective tissue disorders with the genes outlined in Table 1, Table 2 and Table 3.

A limitation of this study is the lack of published scientific literature on refractive surgery outcomes in patients with HDCTs. The creation of these guidelines and recommendations are based on the principles of refractive surgery and corneal biomechanics, knowledge of excimer and femtosecond lasers, known risk factors for post-refractive complications, and the ocular characteristics of each HDCT. Therefore, the guidelines do not serve as definitive cutoffs for eligibility, and surgeons must use their best judgment and intuition to determine the appropriate course of action for each patient. Given the lack of large-scale, long-term studies on refractive surgery outcomes in patients with HDCTs, surgeons need to ensure that patients are fully informed and consented to the high degree of unknown risks and complications before moving forward with these elective surgeries. Nevertheless, the hope is that this paper will change the paradigm that refractive surgery is contraindicated in all patients with HDCTs and prompt future studies, including the role of biomechanical evaluation in determining the safety and efficacy of surgical options in these patients.

## Figures and Tables

**Figure 1 jcm-10-03769-f001:**
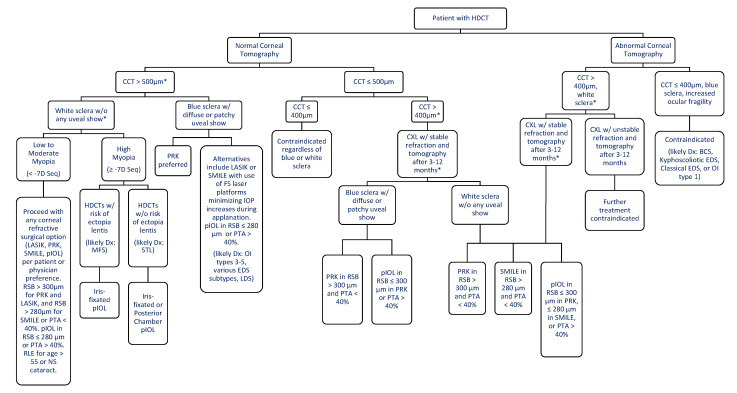
Framework for refractive surgery in HDCT Patients. * Denotes every patient should have the typical biomechanical evaluation when considering refractive surgery measuring corneal hysteresis (CH) and a corneal resistance factor (CRF) using an ocular response analyzer (ORA; Reichert Ophthalmic Instruments, NY, USA) or dynamic schiempflug tonometer (DST; CorVis ST, OCULUS, Wetzlar, Germany). Normal corneal tomography refers to patients without signs of keratoconus, pellucid marginal degeneration or forme fruste keratoconus, asymmetric astigmatism, asymmetric corneal steepening. HDCT = Heritable Disorder of Connective Tissue, CCT—Central Corneal Thickness, CXL = Collagen Cross-linking, LASIK = Laser-Assisted in-situ keratomileusis, PRK = Photorefractive Keratectomy, SMILE = Small Incision Lenticule Extraction, Seq = Spherical equivalent, BCS = Brittle Cornea Syndrome, EDS = Ehlers Danlos Syndrome, OI = Osteogenesis Imperfecta, IOP = Intraocular Pressure, FS = Femtosecond, RSB = Residual Stromal Bed, pIOL = Phakic Intraocular Lens, PTA = Percent Tissue Alteration, RLE = Refractive Lens Exchange, NS= Nuclear Sclerotic.

**Table 1 jcm-10-03769-t001:** Osteogenesis Imperfecta subtypes.

OI Type Prevalence: 1 in 15,000 [22]	Inheritance Pattern	Gene	Severity	Phenotype	Ocular Manifestations	Refractive Surgery Recommendations
1 [4,5,21,23,24,25]	AD	COL1A1/2	Mild	Osteoporosis, fractures, conductive deafness, mild stunting, +/− dentinogenesis imperfecta	Blue sclera, ocular rigidity, significantly lower CCT, corneal hysteresis, corneal resistance factor, global rupture, absence of Bowman’s Layer, glaucoma	Follow the framework in Figure 1 for patients with CCT > 400 µm. Surgery is contraindicated in patients with blue sclera and CCT ≤ 400 µm.
	X-linked	PLS3				
2 [4,5,23,25]	AD (dominant-negative inheritance), AR	COL1A1/2, CRTAP, LEPREI1, PPIB, BMP1	Severe (perinatal lethal form)	Accordion femur, delayed skull ossification, blue sclera	N/A	N/A
3 [4,5,21,23,25]	AD (dominant-negative inheritance), AR	COL1A1/2, CRTAP, LEPREI1, PPIB, FKBP10, SERPINH1, SERINF1, WNT1	Severe	Moderate to severe bone fragility, coxa vara, multiple fractures, marked long bone deformities, early-onset scoliosis, triangular facies, frontal bossing, extreme short stature	Blue sclera in infancy → white sclera in adolescence, absence of Bowman’s Layer, thinner CCT	Select the most appropriate refractive surgery if the minimum criteria are fulfilled according to the framework in Figure 1. Use femtosecond with lowest IOP increase (e.g., Visumax) for LASIK or SMILE procedures.
4 [4,5,23,25]	AD, AR	COL1A1/2, CRTAP, FKBP10, SP7, SERPINF1, WNT1, TMEM38B	Moderate	Moderate to severe bone fragility, deformity of long bones and spinal column, moderate to severe growth stunting, +/− dentinogenesis imperfecta	+/− Blue sclera (rare), thinner CCT	Select the most appropriate refractive surgery if the minimum criteria are fulfilled according to the framework in Figure 1. Use femtosecond with lowest IOP increase (e.g., Visumax) for LASIK or SMILE procedures.
5 [4,5,23,25]	AD	IFTM5	Mild to Moderate	Calcification of interosseous membrane, hypertrophic callus	+/− Blue sclera (rare)	Select the most appropriate refractive surgery if the minimum criteria are fulfilled according to the framework in Figure 1. Use femtosecond with lowest IOP increase (e.g., Visumax) for LASIK or SMILE procedures.

AD = Autosomal dominant, AR = Autosomal recessive, CCT = central corneal thickness, LASIK = laser assisted in situ keratomileusis, PRK = Photorefractive keratectomy, SMILE = Small Incision Lenticule Extraction, IOP = Intraocular pressure.

**Table 2 jcm-10-03769-t002:** Subtypes of Ehlers Danlos syndrome.

EDS Subtype Prevalence: 1 in 5000 [33]	Inheritance Pattern	Gene	Phenotype	Ocular Manifestations	Refractive Surgery Recommendations
Classical EDS [26,37]	AD	COL5A1, COL1A1	Skin hyperextensibility, atrophic scarring, generalized joint hypermobility	Blue sclera, epicanthal folds, ptosis, deep-set eyes, myopia, decreased CCT, steep cornea, conjunctivochalasis	Follow the framework in Figure 1 for patients with CCT > 400 µm. Surgery is contraindicated in patients with blue sclera and CCT ≤ 400 µm.
Classical-like EDS [6,26]	AR	TNXB	Skin hyperextensibility, no atrophic scarring, easy bruising, and generalized joint hypermobility +/− recurrent dislocations	Strabismus, subconjunctival hemorrhage has been reported, astigmatism	Select the most appropriate refractive surgery if the minimum criteria are fulfilled according to the framework in Figure 1. Use femtosecond with lowest IOP increase (e.g., Visumax) for LASIK or SMILE procedures.
Cardiac-valvular EDS [6,26]	AR	COL1A2	Progressive cardiac-valvular problems, skin hyperextensibility, joint hypermobility	Myopia, +/− blue sclera, astigmatism	Select the most appropriate refractive surgery if the minimum criteria are fulfilled according to the framework in Figure 1. Use femtosecond with lowest IOP increase (e.g., Visumax) for LASIK or SMILE procedures.
Vascular EDS [6,26]	AD	COL3A1, COL1A1	Arterial ruptures, sigmoid colon perforations, uterine ruptures, and carotid-cavernous sinus fistulas	Globe protrusion, decreased CCT, rare cases reported of keratoconus	Follow the framework in Figure 1 for patients with CCT > 400 µm. Surgery is contraindicated in patients with blue sclera and CCT ≤ 400 µm.
Hypermobile EDS [6,26,38]	AD	unknown	Generalized joint hypermobility, skin hyperextensibility, bilateral piezogenic papules of the heel, abdominal hernias, atrophic scarring, pelvic organ prolapse, aortic root dilation, mitral valve prolapse	Dry eyes, steep cornea, myopia, vitreous abnormalities	Select the most appropriate refractive surgery if the minimum criteria are fulfilled according to the framework in Figure 1. Consultation with retina before SMILE or PRK. Dry eyes and a steep cornea may contraindicate for LASIK but can be considered for pIOL if there is high myopia
Arthrochalasia EDS [6,26]	AD	COL1A1, COL1A2	Congenital hip dislocations, severe generalized joint hypermobility, skin hyperextensibility	+/− Blue sclera, +/− lens dislocation	Select the most appropriate refractive surgery if the minimum criteria are fulfilled according to the framework in Figure 1. Use femtosecond with lowest IOP increase (e.g., Visumax) for LASIK or SMILE. PRK for those with blue sclera. An iris fixated pIOL can be used for those with lens dislocation.
Dermatosparaxis EDS [6,26]	AR	ADAMTS2	Skin fragility, redundant skin with increased palmar wrinkling, short limbs, severe bruisability	Congenital or early progressive myopia, glaucoma	Select the most appropriate refractive surgery if the minimum criteria are fulfilled according to the framework in Figure 1. Use femtosecond with lowest IOP increase (e.g., Visumax) for LASIK or SMILE. An alternative is pIOL for those with high myopia.
Kyphoscoliotic EDS [6,26,39]	AR	PLOD1, KBP14	Muscle hypotonia, kyphoscoliosis, generalized joint hypermobility	Ocular fragility, keratoconus, blue sclera, myopia, microcornea, decreased CCT, absent Bowman’s membrane	Refractive surgery is contraindicated
Brittle Cornea Syndrome [6,26,35,36,40]	AR	ZNF469, PRDM5	Joint hypermotility, kyphoscoliosis, hyperlaxity of the skin, +/− red hair, conductive hearing loss	Ocular fragility, blue sclera, keratoconus and keratoglobus, megalocornea, myopia, very reduced CCT	Refractive surgery is contraindicated
Spondylodysplastic EDS [6,26]	AR	B4GALT7, B3GALT6, SLC39A13	Short stature, muscle hypotonia, bowed limbs	Hypermetropia, strabismus, corneal clouding, microcornea, glaucoma, refractive errors, +/− blue sclera, astigmatism	Select the most appropriate refractive surgery if the minimum criteria are fulfilled according to the framework in Figure 1. Use femtosecond with lowest IOP increase (e.g., Visumax) for LASIK or SMILE.
Musculocontractural EDS [6,26]	AR	CHST14, DSE	Multiple congenital contractures, skin hyperextensibility, skin fragility with atrophic scars	Myopia, hyperopia, astigmatism, strabismus, microcornea, glaucoma, retinal detachment	Select the most appropriate refractive surgery if the minimum criteria are fulfilled according to the framework in Figure 1. Use femtosecond with lowest IOP increase (e.g., Visumax) for LASIK or SMILE.
Myopathic EDS [6,26]	AD or AR	COL12A1	Congenital muscle hypotonia, proximal joint contractures, hypermobility of distal joints	No reports of ocular manifestations	Select the most appropriate refractive surgery if the minimum criteria are fulfilled according to the framework in Figure 1. Use femtosecond with lowest IOP increase (e.g., Visumax) for LASIK or SMILE.
Periodontal EDS [6,26]	AD	C1R	Severe and intractable periodontitis, lack of attached gingiva, pretibial plaques	No reports of ocular manifestations	Select the most appropriate refractive surgery if the minimum criteria are fulfilled according to the framework in Figure 1. Use femtosecond with lowest IOP increase (e.g., Visumax) for LASIK or SMILE.

AD = autosomal dominant, AR = autosomal recessive, CCT = central corneal thickness, LASIK = laser assisted in situ keratomileusis, PRK = photorefractive keratectomy, SMILE = small incision lenticule extraction, pIOL = phakic intraocular lens, IOP = intraocular pressure.

**Table 3 jcm-10-03769-t003:** Heritable Diseases of Connective Tissue.

HDCT (Subtypes) * Prevalence	Inheritance Pattern	Gene (Warman)	Phenotype	Ocular Manifestations	Refractive Surgery Recommendations
Marfan Syndrome [8,49] * 1 in 5–10,0000 [48]	AD	Fibrillin-1	Long bone overgrowth, aortic root aneurysm, hypermobility, low bone mineral density, scoliosis, pectus excavatum and carinatum, dural ectasia, foot deformities, generalized ligamentous laxity	Ectopia lentis, flattened cornea, increased axial length, astigmatism, hypoplastic iris or hypoplastic ciliary muscle, uveitis, myopia, decreased CCT	Select the most appropriate refractive surgery if the minimum criteria are fulfilled according to the framework in Figure 1. Use femtosecond with lowest IOP increase (e.g., Visumax) for LASIK or SMILE. For those with high myopia, an iris-fixated pIOL can be considered except for those with iridodonesis.
Loeys-Dietz Syndrome [10,50] (Types 1–5) * <1 in 100,000 [51]	AD	TGFBR1, TGFBR2, SMAD3, TGFB2 or TGFB3	Vascular aneurysms of the cerebral, thoracic, and abdominal arterial system, skeletal abnormalities such as pectus excavatum or carinatum, scoliosis, joint laxity and craniofacial abnormalities such as cleft palate, craniosynostosis, and bifid uvula, No lens dislocation	Blue or dusky sclera, hypertelorism, myopia, cataracts, retinal detachment, retinal tortuosity, strabismus, decreased CCT	Select the most appropriate refractive surgery if the minimum criteria are fulfilled according to the framework in Figure 1. Use femtosecond with lowest IOP increase (e.g., Visumax) for LASIK or SMILE. For blue and dusky sclera PRK can be preferred.
Epidermolysis Bullosa [12] (Main Types: EB Simplex, Junctional EB, Dystrophic EB and Kindler Syndrome) * 0.4–4.6 per million people [12]	AD	keratin 5 or 14, Laminin-322, COLA71 or FERMT1 (KIND1)	Epithelial tissue fragility, resulting in blisters and erosions from minimal trauma	Corneal erosions, conjunctival injections blistering of eyelids, pannus formation, symblepharon, corneal scarring	Select the most appropriate refractive surgery if the minimum criteria are fulfilled according to the framework in Figure 1. Patients with dry eye and/or optimized ocular surface PRK or SMILE would be better options. Check limbal stem cells with impression cytology or high-res AS-OCT. Patients with uncontrolled ocular surface disease can be considered for pIOL.
Stickler Syndrome [13,52] (Types 1–6) * 1–3 in 10,000 [53]	AD	COL2A1, COL11A1, COL11A2, COL9A1, COL9A2, COL9A3, LOXL3. COL2A1, COL11A1, or COL11A2	Conductive and sensorineural hearing loss, midfacial underdevelopment and cleft palate, spondyloepiphyseal dysplasia	High myopia, cataracts, glaucoma, retinal detachments, vitreous abnormalities	Retina consultation before selecting the most appropriate refractive surgery if the minimum criteria are fulfilled according to the framework in Figure 1. Use femtosecond with lowest IOP increase (e.g., Visumax) for LASIK or SMILE. The use of pIOL can be considered for patients with high myopia.
	AR	COL9A1, COL9A2, COL9A3, or LOXL3			
Wagner Syndrome [54,55] * 300 affected worldwide [55]	AD	VCAN	No systemic abnormalities	Empty vitreous, chorioretinal atrophy, myopia, night blindness, retinal detachment, presenile cataract, uveitis	Retina consultation before selecting the most appropriate refractive surgery if the minimum criteria are fulfilled according to the framework in Figure 1. Use femtosecond with lowest IOP increase (e.g., Visumax) for LASIK or SMILE. The use of pIOL can be considered for patients with high myopia.
Pseudoxanthoma Elasticum [17,56,57] * 1 in 25–100,000 [56]	AR	ABCC6	Accumulation of elastic fiber in the skin, vasculature and, Bruch’s membrane of the eye	Angioid streaks, choroidal neovascularization, peau d’orange appearance on fundoscopy, optic disc drusen, choroidal atrophy with comet tails, submacular hemorrhage	Retina consultation before selecting the most appropriate refractive surgery if the minimum criteria are fulfilled according to the framework in Figure 1. Use femtosecond with lowest IOP increase (e.g., Visumax) for LASIK or SMILE.

* Denotes prevalence. AD = autosomal dominant, AR = autosomal recessive, CCT = central corneal thickness, LASIK = laser assisted in situ keratomileusis, PRK = photorefractive keratectomy, CXL = collagen cross linking, SMILE = small incision lenticule extraction, High-res AS-OCT = high resolution anterior segment optical coherence tomography, pIOL = Phakic intraocular lens.

## Data Availability

We confirm that this publication is original.

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
