# Peer review of "Controversy and Consideration of Refractive Surgery in Patients with Heritable Disorders of Connective Tissue"

_jcm, 2021, doi:10.3390/jcm10173769_

Round 1

Reviewer 1 Report

In this paper the authors present a review of refractive surgery in patients with collagen diseases. This is an interesting topic, and the authors do a nice job of collecting and summarizing the available evidence.  This represents a real "hot" topic in the ophthalmology field. I find the manuscript overall well-structured and accurate. Reading is fluent and numerous accuracies denote precision. 

Can the authors provide a merged figure with some of the most common corneal collagen diseases. Secondly I wonder if the authors can write a section about presbyopic refractive procedures in patients with collageneses. 

A city where Carl Zeiss Meditec is located should be named. 

Reviewer 2 Report

This is an interesting review paper looking at refractive surgery in the setting of a connective tissue disorder. The manuscript is interesting  but has several potential limitations:

Change keratoconus to underlying keratoconus in abstract

Risk of globe rupture with LASIK in connective tissue disease needs to be clarified in the introduction.

Marfan: For those with high myopia, an iris-fixated pIOL can be considered, what is the evidence of this given the degree of potential iridodonesis present in such patients.

It is worth beefily discussing tissue sparing techniques in ablative procedures such as reducing optical zone etc.

Figure 1 has a number of potential issues:

No mention of biomechanical evaluation or combined tomography and biomechanical if CCT <400um. Evaluation is made.

It is understandable that there is a lack of evidence for many of the suggestions but the authors need to justify the use of SMILE over PRK for patients with an RSB of 280-300.

What is the evidence that SMILE is biomechanically advantageous compared to PRK? This is debatable.

The use of CXL prior to treatment if CCT <400um needs justification.

I would postulate that all patients should be advised to undergo ICL surgery as the first choice if they have no signs of ectopia lentis or LSCD and meet other ICL implantation requirements, which has a much lower risk of associated corneal/biomechanical issues.

Refractive lens exchange is not discussed. 

It should be stressed and highlighted in the conclusion section that the patient consent process should include a discussion on the lack of long-term results and the high degree of uncertainty that any refractive surgery presents in such settings.

Reviewer 3 Report

This is a comprehensive review that will be useful for clinicians in making a decision on refractive index surgery based on the type of connective tissue heritable disorder. Limitations of the study have been noted by the authors. 
